# New Parents Experienced Lower Parenting Self-Efficacy during the COVID-19 Pandemic Lockdown

**DOI:** 10.3390/children8020079

**Published:** 2021-01-24

**Authors:** Anja Xue, Vivian Oros, Pearl La Marca-Ghaemmaghami, Felix Scholkmann, Franziska Righini-Grunder, Giancarlo Natalucci, Tanja Karen, Dirk Bassler, Tanja Restin

**Affiliations:** 1Newborn Research Zurich, Department of Neonatology, University Hospital and University of Zurich, 8091 Zurich, Switzerland; anja.xue@bluewin.ch (A.X.); vivian.oros@bluewin.ch (V.O.); felix.scholkmann@usz.ch (F.S.); giancarlo.natalucci@usz.ch (G.N.); tanja.karen@usz.ch (T.K.); dirk.bassler@usz.ch (D.B.); 2Department of Clinical Psychology and Psychotherapy, University of Zurich, 8050 Zurich, Switzerland; pearl.lamarca@peace-academy-society.org; 3Psychology Counselling and Research Institute for Sexuality, Marriage and the Family, International Academy for Human Sciences and Culture, 8880 Walenstadt, Switzerland; 4Department of Pediatric Gastroenterology, Children’s Hospital Lucerne, 6000 Luzern, Switzerland; franziska.righini@luks.ch; 5Larsson-Rosenquist Center for Neurodevelopment, Growth and Nutrition of the Newborn, University Hospital and University of Zurich, 8952 Zurich, Switzerland; 6Institute of Physiology, University of Zurich, 8057 Zurich, Switzerland

**Keywords:** COVID-19, parental self-efficacy, TOPSE, perinatal period, newborn, depression, EPDS, quality of life, couple satisfaction

## Abstract

*Background*: The COVID-19 pandemic is a global issue which affects the entire population’s mental health. This study evaluates how restrictions to curtail this pandemic change parenting self-efficacy, depressive symptoms, couple satisfaction and health-related quality of life in parents after delivery of a newborn. *Methods*: In this prospective single center evaluation of parental self-efficacy and quality of life, four validated questionnaires were used to repeatedly assess parenting self-efficacy (Tool to measure Parental Self-Efficacy, TOPSE), depressive symptoms (Edinburgh Postnatal Depression Scale, EPDS), couple satisfaction (Couple Satisfaction Index, CSI) and health-related quality of life (short form 12, SF12). Fifty-three parents of 50 infants answered a total number of 63 questionnaires during the lockdown period to limit the spread of COVID-19. These questionnaires were matched with 63 questionnaires of 58 other parents that had answered them before or after strong pandemic related measures. *Results*: Parents experienced lower parenting self-efficacy during the strict pandemic measures as compared to before and after (*p* = 0.04). In terms of age, socioeconomic, marital status and duration of hospitalization we detected no significant difference between both groups. On univariate linear regression, TOPSE scores were associated with gestational age (*p =* 0.044, parameter estimate: 1.67, 95% CI: 0.048 to 3.301), birth weight (*p* = 0.035, parameter estimate: 0.008, 95% CI: 0.001 to 0.015), number of newborns’ siblings (*p* = 0.0554, parameter estimate: 7.49, 95% CI: −0.174 to 15.145) and distance of home from hospital (*p* = 0.043, parameter estimate: −0.38, 95% CI: −0.745 to −0.011). Interestingly, there was a positive correlation between quality of life and TOPSE scores, suggesting that those who experience a higher self-efficacy also have a higher quality of life. *Conclusions*: When implementing a lock-down period psychological effects such as lower experience of parental self-efficacy have to be considered.

## 1. Introduction

The COVID-19 pandemic due to infection with SARS-CoV-2, a coronavirus first described in 2019, affects the world in many different ways. On March 11th 2020 the World Health Organization declared the state of a global pandemic [1]. In Switzerland, the government reacted and took pandemic measures starting from March 17th on by advising people to stay at home, recommending home-office and calling up military service in case of need for more employees in the health system [2]. Correspondingly, since 12 March, no visitors were allowed during the hospital stay at the University Hospital Zurich with the exception that fathers were tolerated in the delivery room during the immediate delivery of their child [3]. We named these social restrictions “pandemic lockdown period”. From 27 March on, some COVID-19 measures were eased [4] and the visiting ban at the University Hospital of Zurich was lifted on 30 May (according to internal communication, USZ). Recent studies showed that the pandemic and the associated social measures had an impact on mental health in the entire population [5,6,7]. It is known from previous pandemics, such as SARS-CoV-1 in 2003, that the population suffers from psychological distress afterwards [8]. In earlier pandemics new parents experienced symptoms of depression and anxiety, as well as stress, due to lots of uncertainties involving perinatal care of their child [9]. Self-efficacy describes the people’s belief in their own abilities to complete a given task [10]. Studies suggest that a higher self-efficacy may correlate with a better task performance [11,12] and lower psychological distress [13]. However, family well-being is influenced by many factors, some external such as social disruption by COVID-19 measures but also altered couple relationships changed due to home-office and home-schooling. Data show that in previous pandemics the couple relationships in families can also improve in some cases as both partners were able to support each other [9,14,15]. A recent study from China has already shown that parents of children hospitalized due to the pandemic had more serious mental health problems than parents of children hospitalized due to medical issues unassociated with the pandemic [16]. In this study, we aim to determine the effect of COVID-19 measures on parents’ self-efficacy, depressive symptoms, couple satisfaction and quality of life.

## 2. Materials and Methods

### 2.1. Study Subjects

During the recruiting period starting in December 2018, parents of term and preterm born newborn infants delivered at the University Hospital Zurich were approached at the maternity ward and neonatology unit. Parents who were able to complete the questionnaire in German or English and willing to participate, were included in the study. Exclusion criteria were parents of children with genetic abnormalities or severe morbidities detected before or after birth (such as congenital heart disease, trisomy 21, 18 and 13).

### 2.2. Procedure

This research project is part of an ongoing single-center prospective study evaluating parental self-efficacy, quality of life, postpartum depressive symptoms and couple relationship in parents of newborn infants at the University Hospital of Zurich since December 2018. During the COVID-19 pandemic the project was allowed to be continued in a restricted way. Consequently, parents were invited to participate, but only if the contact between the physician and parents was needed for clinical reasons. Whenever possible, mother and father were recruited, but it was also possible that one parent participated separately. The analysis took place after the measures that were taken due to COVID-19 pandemic were eased. In this study, we distinguished between strong pandemic measures from 17 March 2020 to 30 May 2020 (lockdown period), characterized by governmental recommendation to stay at home, to work from home, to keep social distancing, and, specific visiting ban at University Hospital of Zurich), and the time before the measures were put in place or after those measures were eased. During the lockdown period 53 parents (43 mothers and 10 fathers) completed 63 questionnaires. These questionnaires were statistically matched by survey point and gestational age of the infant with 63 questionnaires of 58 parents (40 mothers and 18 fathers) that had participated in the study before or after the pandemic measures according to the consort diagram (Figure 1). Mothers and fathers responded to the questionnaires independently and repeatedly at three measuring times: 1) during the first week postpartum, 2) after six weeks postpartum and 3) after three months postpartum. If the baby was born preterm, parents filled in the questionnaire 1) in the first week postpartum, 2) at term equivalent 3) three months postpartum. Some infants were treated in the maternity ward and some were hospitalized at a neonatology unit.

### 2.3. Questionnaire

The entire questionnaire consisted of the tool to measure parenting self-efficacy (TOPSE score), the Edinburgh Postnatal Depression Scale (EPDS), the Couple Satisfaction Index (CSI) and the Short Form 12 (SF-12).

TOPSE is a validated tool to assess parenting self-efficacy [17]. Parents are asked to rate the extent to which they agree to a total of 30 statements regarding parenting self-efficacy on a scale from one to ten. It covers different topics of parenting including both emotions, affection and pressures they feel as a parent as well as self-acceptance and confidence. Parenting self-efficacy is defined as the parents’ own perception about their ability to successfully care for their newborn child. A higher score indicates a better parenting self-efficacy.

The EPDS is a screening tool for symptoms of depression and anxiety. The questionnaire is composed of ten questions about feelings parents experience postpartum. The answers are scored with 0 to 3 points and summed together for the score values. Higher scores indicate higher depressive symptomatology [18]. Although it was originally created to screen for depressive symptoms in mothers, the EPDS has also been validated for fathers [19].

The CSI consists of four questions which evaluate relationship satisfaction in couples [20]. Each question has six answer choices, which are rated with 0 to 5 points and summed together. Higher scores indicate more couple satisfaction.

The SF-12 is a questionnaire validated to measure health-related quality of life [21]. It consists of twelve questions divided in eight dimensions: physical functioning, physical role, bodily pain, general health, vitality, social functioning, emotional role and mental health. For each dimension a score from 0–100 points is calculated. SF-12 total score is the mean value of the eight dimensions. Sub-scores for physical and mental health can be analyzed separately. A higher SF-12 score indicates a better health-related quality of life. CSI, SF-12 and TOPSE questionnaires were not specifically validated for women only and can be applied for both women and men [17,21,22].

### 2.4. Primary and Secondary Outcomes

The primary endpoint was the investigation for parental self-efficacy (TOPSE score) and the influence of the lockdown period. Secondary outcomes were the investigation for health-related quality of life (total scores of SF-12), the evaluation of EPDS and of the CSI questionnaires and investigation of variables that might have an impact on parental self-efficacy (TOPSE score) such as gestational age, birth weight, gender, proxy filling in the questionnaire (mother vs. father), family socioeconomic status, age and health status of parents, parental self-estimation of quality of life, number of the newborns’ siblings and distance from hospital to home.

### 2.5. Statistics

The questionnaires completed during the lockdown period were statistically matched to controls before or after the lockdown period by survey times (1 week after birth, 6 weeks after birth (at term for preterm birth), and 3 months after birth) and by gestational age of the participants’ infants. The data were processed and analyzed using GraphPad Prism version 8.0 (GraphPad Software, San Diego, California USA), SAS, version 9.3 (SAS Institute, Cary, NC, USA), and scistat (MedCalc Software Ltd., Ostend, Belgium) and JASP (JASP Team, v. 0.11.1). Visualization of the results was performed with R (version 4.0.2). The data was tested for normal distribution. All statistical tests were two-sided. *p* values < 0.05 were considered statistically significant for all analyses. Wilcoxon rank sum test (Mann Whitney U test) was used to analyze the influence of strong pandemic measures on univariate analysis. Additionally, a linear multivariable regression model with the TOPSE Score as outcome variable and strong pandemic measures as principal exposure variable was carried out, including variables with a significance on univariate analysis at ≤0.02 or clinical importance.

### 2.6. Ethics

The responsible local ethics committee approved the research plan on 29.11.2018 (Nr: 2018-01796). Participating parents signed an Informed Consent.

## 3. Results

### 3.1. Characteristics of the Study Population

From December 2018 until June 2020, 289 parents of 160 term and 66 preterm children were included in the research project. The response rate was 42%. 63 questionnaires, completed by 53 participants during the lockdown period were statistically matched with 63 questionnaires, completed by 58 participants prior or post the lockdown period (55 questionnaires filled in prior and eight questionnaires filled in after the lockdown period). 36 parents answered one questionnaire during the lockdown period, 10 parents answered two questionnaires during the lock-down period and seven parents answered questionnaires both during and before or after the lockdown period. We had 11 couples participating which means mother and father answered the questionnaires separately. Eight of these couples participated during lockdown period, one couple before lockdown period and of two couples mother and father answered the questionnaire with a time difference of a few days, which caused them to be in both periods (before/after and during lockdown period). Altogether, 64 questionnaires were evaluated for the first point of time of the survey (1 week after birth), 42 questionnaires for the second point of time (at predicted term or 6 weeks after birth) and 20 questionnaires at the third point of time (3 months after birth). During the time before the strong pandemic measures three parents answered more than one questionnaire, during “lockdown” measures ten answered two questionnaires. The characteristics of the study population are shown in Table 1. Timeline of filling in the questionnaires is depicted in Figure 2.

### 3.2. Primary Outcome During COVID-19 Lockdown Period

On univariate analysis, when comparing the questionnaires completed during the strong COVID-19 pandemic measures to those completed before or after the lockdown situation, parenting self-efficacy (TOPSE score) was significantly lower (*p* = 0.041, effect size: 0.215) (shown in Table 2 and Figure 3. When performing a subanalysis of the matched data, excluding questionnaires of the seven parents that answered prior/after and during lockdown period and their matched questionnaires, TOPSE is also significantly lower (*p* = 0.008, 95% CI: −0.752 to 0.058, effect size: −0.347). When performing the same subanalysis on the matched data excluding questionnaire of the eleven fathers of the participating couples and their matched questionnaires, TOPSE is still significantly lower (*p* = 0.048, effect size −0.309). A combined subanalysis without individuals filling out several questionnaires in both groups and excluding fathers when couples filled out questionnaires and all of their matching questionnaires, TOPSE during lockdown period is significantly lower (*p* = 0.011, effect size: −0.479). Overall matched questionnaires when comparing all TOPSE scores of mothers and fathers in a univariate analysis there was no significant difference found (*p* = 0.183, effect size: 0.262).

### 3.3. Secondary Outcomes during COVID-19 Lockdown Period

EPDS, CSI as well as SF-12 total and subscores did not show any difference in the two groups before or after and during the COVID-19 pandemic lockdown situation (Table 2). No significant differences between questionnaires filled in by mothers compared to fathers were observed.

Predictors of parenting self-efficacy (TOPSE score) were analyzed using linear regression analysis. On simple linear regression, a significant association between TOPSE score and gestational age (*p* = 0.044, parameter estimate: 1.67, 95% CI: 0.048 to 3.301), birth weight (*p* = 0.035, coefficient: 0.008, 95% CI: 0.001 to 0.015), number of newborns’ siblings (*p* = 0.055, parameter estimate: 7.49, 95% CI: −0.174 to 15.145) and distance of home from hospital (*p* = 0.043, parameter estimate: −0.38, 95% CI: −0.745 to −0.011) was found. The linear multivariable regression analysis with TOPSE scores as outcome variable and strong pandemic measure (lockdown period) as principal exposure variable, adjusted for gestational age, birth weight, gender and including clinical relevant variables and/or significance at ≤ 0.2 results in univariate analysis (parental filling in the questionnaire (mother vs. father), age of parent at time of filling in the questionnaire, presence of physical or psychological parental problems, number of siblings, parental self-estimation of quality of life and socioeconomic status) was performed. The lockdown period was negatively associated with TOPSE scores, indicating that parents reported lower self-efficacy during strong pandemic measures (*p* = 0.0497, parameter estimate: −11.5, 95% CI: −23.04 to −0.014). Assessing the raw data, we found one significant outlier with a TOPSE score below 100 in the group of parents filling in the questionnaire before the lockdown period. Statistically, due to the difference of more than 2 standard-deviations to the mean, it might be appropriate to exclude this value which then would result in even more significant results with a significant reduction of TOPSE during the lockdown period (*p* = 0.013). In order to not overestimate the effects, we cautiously kept the value within the dataset. Parental self-assessment of quality of life was positively associated with higher TOPSE scores (*p* = 0.0297, parameter estimate: 4.549, 95% CI: 0.457 to 8.641).

## 4. Discussion

The objective of this research was to evaluate the influence of the lockdown period on parental self-efficacy after delivery of a child at the University Hospital Zurich. It became clear that parents who cared for an infant younger than 3 months during the lockdown period because of the COVID-19 pandemic exhibited less parenting self-efficacy compared to those parents, who were in the same situation before or after pandemic measures. This result was shown to be robust, no matter whether we kept or excluded serial questionnaires or both questionnaires filled out separately by fathers or mothers belonging to the same couple and child/children. On a professional level, increased self-efficacy is correlated with a higher level of job-satisfaction and job-performance, especially concerning tasks of a lower complexity [23]. Increasing evidence suggests that parental self-efficacy may also improve parental competences as reviewed by Jones [24]. These findings are in line with the study of Leahy-Warren and Mc Carthy who found that maternal self-efficacy decreases with stress, anxiety and depression while it increases with social support and parenting satisfaction [25]. To our opinion, the lockdown period was a completely new situation with new uncertainties for parents and the caring personnel which made personnel interaction and social support more difficult. These facts may have contributed to the reduction of self-effectiveness. Additional studies suggest that pandemic measures may be associated with increased levels of depression and anxiety [26] and may even correlate with long-term psychological consequences such as posttraumatic stress disorder [27,28]. In Switzerland the “Swiss Corona Stress Study” demonstrates that about 40% of responders felt more stressed during the pandemic compared to the time before [29] and in Italy, anxiety, perceived stress and adjustment disorder affected about every 5th person who filled in the questionnaire during the Italian lockdown period [30]. Interventions which strengthen parental self-efficacy such as family centered or family integrated care [31,32] should therefore be promoted during times of insecurity such as a pandemic. Interestingly, our secondary analysis demonstrated a positive correlation between self-assessment of quality of life and TOPSE scores, suggesting that those with higher self-efficacy also experience a better quality of life. According to Banduras theory main sources of self-efficacy are the observation of role models, the opportunity to practice, receiving feedback and the affective state [33]. During the lockdown period the social contacts have been reduced, which may have led to a reduction of potential role models, a reduction of opportunities to practice with respective feedbacks of peers and/or personnel. Interestingly, parents with preterm children and those with children at a lower birth weight also showed a reduced self-efficacy compared to parents with term children of higher weight. This result may be partially due to the reduced emotional feedback of preterm children and possibly increased concerns and uncertainty due to prematurity and reduced weight [34].

However, in contrast to previous studies in this study population of new parents’ the self-reported quality of life as well as symptoms of depression and anxiety did not differ due to pandemic measures [5,6,16,35]. It may be that the low self-efficacy we detected using the TOPSE score may later on predispose for depression or adjustment problems as suggested by the cognitive vulnerability hypothesis [36,37]. Perhaps the assessment time in our setting was too early to detect these problems. The fact that fathers were allowed to stay in the hospital for deliveries only and could not visit their child and partner afterwards could be one reason for lower self-efficacy of both fathers and mothers in our study. In their responses, mothers reported about uncertainties before delivery being frightened how the delivery would take place and if their partner could arrive on time. Also, the psychological well-being of the health care team might have been affected by the COVID-19 measures because of a different interaction with their patients [38,39,40,41]. The design of the study restricts the cases to the time of the “lockdown period”, not taking into account that a certain fear and anxiety concerning the spreading pandemic may both already occur before the implementation of strong measures and may persist after lockdown release. In Zurich, unlike in other cities, there has not been a relevant pandemic related shortage of medical resources, and the medical support after delivery by both nurses and midwifes has always been available. Studies describing the association of depression following quarantine mainly involve people who were advised to completely stay at home [42] and were assessed years later [43], while in Zurich it was still allowed to go outside and meet in groups ≤ 5 persons. Additionally, it is possible that some parents could enjoy more family time at home due to parental working in home office which may have helped to prevent mental health issues. Another possible reason could be that pandemic measures in Switzerland were not as strict as compared to China or Denmark. The Danish study of Sønderskov et al. found that the general population was negatively affected by the pandemic, with women being more affected than men [6]. However, our study could not confirm this difference in our population investigated. Parenting self-efficacy is important because it affects children’s upbringing [44,45].

This study has several limitations. It was not assessed how much time fathers could spend with their family after the delivery of the newborn. In Switzerland, fathers officially get one day off at work after the birth of their child, but some companies offer more days of paternity leave or some fathers take their holidays to be with their family. The control group consists of individuals filling in the questionnaire before OR after the lockdown period and some individuals filled in a questionnaire at several times, couples participating, resulting in a heterogenous group with overlaps, which could influence the results. However, to control for this potential bias, we matched the questionnaires with controls by survey point and gestational age. Subanalyses showed that couples or individuals filling out questionnaires at several times did not influence the significant result for lower TOPSE scores during lockdown period. Furthermore, we controlled the potential bias factors by performing the regression analysis on the matched data, including the same parameters to double check for any influence on the primary outcome TOPSE Score.

## 5. Conclusions

This study demonstrates a significant lower parental self-efficacy during the phase when the COVID-19 pandemic measures in Switzerland were active, compared to the situation before or after. Health care teams should consider intervention programs to support parenting self-efficacy, especially during situations of global uncertainty.

## Figures and Tables

**Figure 1 children-08-00079-f001:**
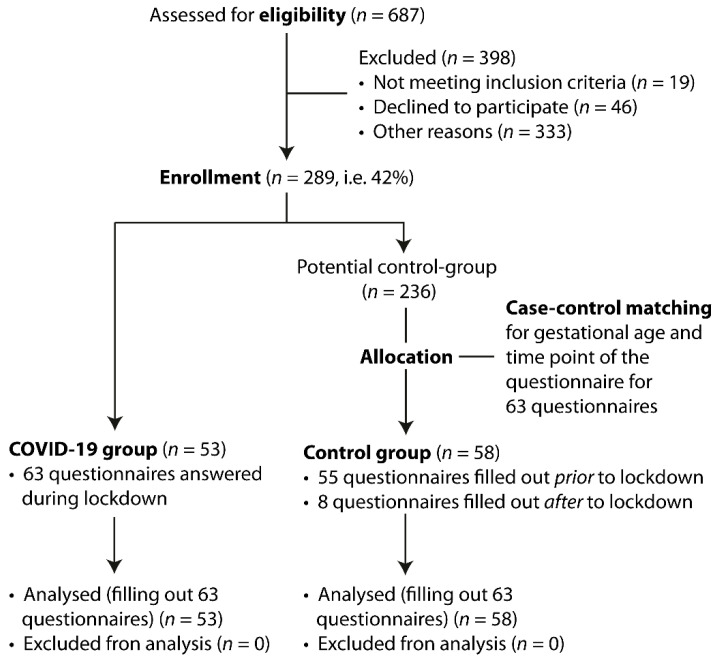
CONSORT diagram for enrollment and study allocation.

**Figure 2 children-08-00079-f002:**
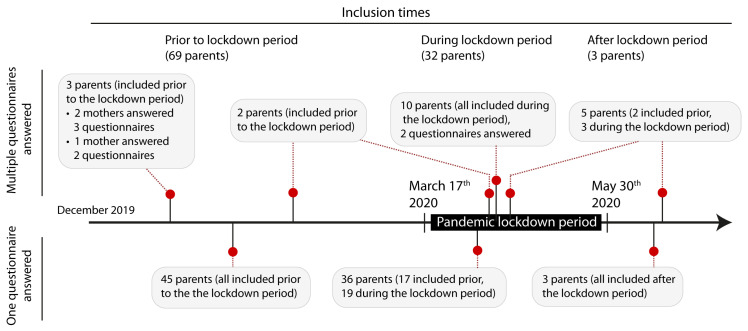
The lockdown period interfered unexpectedly with the study evaluation of parental self-efficacy, that is why the questionnaires filled in during the lockdown period were statistically matched with questionnaires of parents with children of the same gestational age and the same time after delivery.

**Figure 3 children-08-00079-f003:**
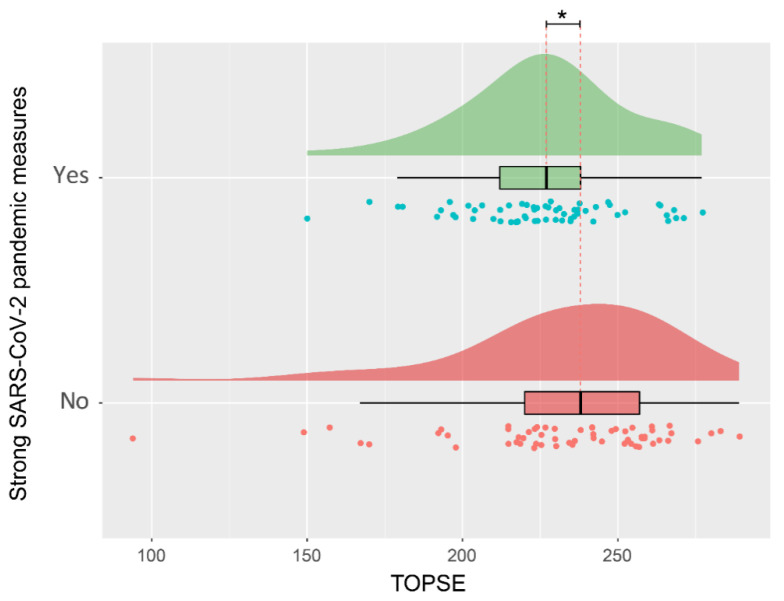
Visualization of the distribution of parental self-efficacy during the lock-down period or before and after. * Significant difference *p* < 0.05.

**Table 1 children-08-00079-t001:** Baseline characteristics of 104 parents and 103 children.

Characteristics	During Lockdown	Before/After Lockdown	*p* Value
**Parents**			
Participating parents	53	58	
Mothers	43 (81%)	40 (69%)	0.14
Fathers	10 (19%)	18 (31%)	
Participating couples (both parents filled in the questionnaire)	10	1	
Age (years)	35.8 ± 4.9	35.4 ± 5	0.584
Marital status			
Married	40 (75%)	40 (69%)	0.345
Unmarried	10 (19%)	15 (26%)	
Divorced	2 (4%)	2 (3%)	
Registered partnership	1 (2%)	1 (2%)	
Socioeconomic status	2.9 ± 1.5	3.1 ± 1.7	0.515
Duration of prepartum hospitalization (range) in days	2.3 (0–24)	2.4 (0–39)	0.480
Previous medical conditions	13 (26%)	9 (16%)	0.083
Physical	13 (25%)	6 (10%)	
Mental	1 (2%)	3 (5%)	
Taking medication	11 (21%)	5 (9%)	
Attending psychotherapy	3 (6%)	3 (5%)	
Previous children			
0	36 (68%)	43 (74%)	0.35
1	13 (25%)	8 (14%)	
>1	4 (7%)	7 (12%)	
**Infants**			
Newborns	50	61	
Male	26 (52%)	34 (56%)	0.57
Female	24 (48%)	27 (44%)	
Preterm infants	19 (38%)	24 (39%)	0.884
Gestational age (range) (weeks)	37 2/7 (25 4/7–40 4/7)	37 0/7 (25 5/7–41 1/7)	0.62
Birthweight (grams)	2760 ± 826	2723 ± 794	0.888
Multiple gestation resp. twins	5	4	
Conception			
Natural	39 (78%)	53 (87%)	0.094
In Vitro Fertilization	5 (10%)	4 (7%)	
Intra Cytoplasmic Sperm Injection	6 (12%)	4 (7%)	
Mode of birth			
Vaginal delivery	22 (44%)	19 (31%)	0.057
Operative assisted	2 (4%)	6 (10%)	
Caesarean section	26 (52%)	36 (59%)	
Duration of neonatal hospitalization (range) in days	9 (2–73)	11 (2–88)	0.082

Data are given as absolute counts (percentages %) or means ±standard deviation, respectively. Categorial variables were analyzed with χ2 test while continuous variables were analyzed with Wilcoxon rank sum test. Note: 7 parents and 8 children are part of both groups.

**Table 2 children-08-00079-t002:** Characteristics of standardized evaluation of tendency to depression (EPDS), couple satisfaction (CSI), quality of life (SF-12) and self-efficacy (TOPSE). A *p*-value of <0.05 is considered significant meaning that parental self-efficacy has been reduced during the lockdown period (marked in bold). Values given as mean (95% CI).

Scores	Lockdown Period	Control Group (Before/After Lockdown Period)	*p*-Values	Effect Sizes
	*n* = 63	*n* = 63		
EPDS	4.0 (2.0, 8.0)	5.0 (2.0, 8.0)	0.823	0.023
CSI	19.0 (15.5, 20.0)	19.0 (16.0, 20.0)	0.863	0.017
SF-12 total	78.1 (67.7, 87.2)	80.9 (77.8, 90.3)	0.361	0.097
SF-12 physical	81.2 (62.2, 93.1)	88.4 (76.7, 93.8)	0.123	0.161
SF-12 mental	78.1 (70.3, 85.9)	78.1 (65.6, 84.4)	0.532	−0.065
TOPSE	227.0 (212.0, 238.0)	238.0 (220.0, 257.0)	**0.041**	**0.215**

## Data Availability

Not applicable.

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
