# Peer review of "New Parents Experienced Lower Parenting Self-Efficacy during the COVID-19 Pandemic Lockdown"

_children, 2021, doi:10.3390/children8020079_

Round 1
Reviewer 1 Report
The manuscript presents the results of a study that evaluated the change in parental self-efficacy, depressive symptoms, couple satisfaction, and HRQoL in parents after delivery of a newborn during the restrictions due to the pandemic of COVID. The study is based on the responses to 4 questionnaires that are well described and administered before, during, or after the strong restrictions of the lockdown period.
The manuscript is well organized, well written, using correct and fluent English. There are some minor improvements:
- in the abstract, there is no need for numbers (1), (2),... before paragraphs. Also, there should be more results presented.
- in the introduction line 36 world health organization should be with the first capital letter W... H... O...
- line 76 in Procedure there is a repeated (lockdown period)
- in Table 1 - the head of the table that appears a second time should be erased; abbreviations used should be explained in the legend (IVF and ICSI); the title of the table should be before the table.
Author Response
Thank you very much for your thorough review and feedback.

Reviewer 2 Report
In general the manuscript" New Parents Experienced Lower Parenting Self Efficacy during the COVID-19 Pandemic Lockdown" is good. The topics is really hot and extremely actual.
I recommend a few modification.
Double check the English.
Please describe the abbreviations when you use them for the first time
In the discussion part you need to compare with other studies from your country or surrounding region.
The conclusion need to clear and specific.
My recommendation is to focus on short conclusion.
Thank you again for the opportunity to review this interesting manuscript.
Author Response
Thank you very much for your review.
